# Telepsychology in Europe since COVID-19: How to Foster Social Telepresence?

**DOI:** 10.3390/jcm12062147

**Published:** 2023-03-09

**Authors:** Lise Haddouk, Carine Milcent, Benoît Schneider, Tom Van Daele, Nele A. J. De Witte

**Affiliations:** 1Centre Borelli (UMR9010), Rouen University, 76000 Rouen, France; 2CNRS, Paris School of Economics, 75014 Paris, France; 32LPN, Université de Lorraine, 56532 Nancy, France; 4Expertise Unit Psychology, Technology and Society, Thomas More University of Applied Sciences, 2018 Antwerp, Belgium

**Keywords:** telepsychology, social telepresence, mental healthcare professionals, training, acceptance

## Abstract

All over the world, measures were taken to prevent the spread of COVID-19. Social distancing not only had a strong influence on mental health, but also on the organization of care systems. It changed existing practices, as we had to rapidly move from face-to-face contact to remote contact with patients. These changes have prompted research into the attitudes of mental healthcare professionals towards telepsychology. Several factors affect these attitudes: at the institutional and organizational level, but also the collective and personal experience of practitioners. This paper is based on an original European survey conducted by the EFPA (European Federation of Psychologists’ Associations) Project Group on eHealth in 2020, which allowed to observe the variability in perceptions of telepsychology between countries and mental healthcare professionals. This study highlights different variables that contributed to the development of attitudes, such as motivations, acquired experience, or training. We found the “feeling of telepresence”—which consists of forgetting to some extent that we are at a distance, in feeling together—and social telepresence in particular as main determinants of the perception and the practice of telepsychology.

## 1. Introduction

Worldwide, actions were taken to avoid the spread of the COVID-19 pandemic caused by SARS-CoV-2. Aside from reducing the virus’ expansion, it also led to physical distancing, travel restrictions, and city-wide quarantines [1]. The pandemic context constrained people to being isolated, unemployed, inactive, and with low social support which increased patients’ exposure to factors known to exacerbate mental health disorders [2]. Even more so, all of these measures had a tremendous impact on the delivery of healthcare services, including mental healthcare: consulting patients in person became very challenging [3,4,5]. In addition, under strict infection control, some clinical psychiatrists refused to enter COVID-19 patients’ wards [6]. As a result, innovative solutions were needed to address the critical needs of patients. Since the beginning of the pandemic, many healthcare services therefore shifted from in-person to remote patient contact. According to Patel et al. [7], the COVID-19 pandemic has been associated with a marked increase in remote mental health consultation. Telepsychology practices have also increased significantly in the US following the COVID-19 pandemic [8].

For over two decades, research has been carried out, particularly in North America, on the effectiveness of telepsychology services for patients with different types of mental disorders [9,10,11,12]. Telepsychology has been defined by the American Psychological Association (APA) in 2013 as: “the provision of psychological services using telecommunication technologies” (https://www.apa.org/practice/guidelines/telepsychology, accessed on 25 September 2022). The pandemic has prompted a sharp increase in telepsychology practice by new practitioners: a recent study among 1791 mental healthcare professionals in the UK revealed a 12-fold increase in the use of telepsychology [13]. A survey in France, conducted in 2021 among 511 psychologists, showed very infrequent use of telepsychology before the crisis (7.1% of psychologists), whereas more than 70% of psychologists used it during the crisis; slightly less than half (45.6%) estimated that they would still use it after the crisis [14]. This development has prompted new research that shows a gradual acceptance of telepsychology [15]. Prior to the COVID-19 pandemic, the use of teleconsultations in psychiatry and psychology was quite limited in Europe, although these practices existed in some hospitals. For example, in France, telepsychiatry practices, i.e., the application of telemedicine to the specialized field of psychiatry, have been regulated by the decree on telemedicine produced in 2010 (https://www.legifrance.gouv.fr/loda/id/JORFTEXT000022932449, accessed on 25 September 2022). On telepsychology and the use of technology in mental healthcare, some guidelines have in the meantime been produced in Europe [16], but not on a large scale. 

A key concept in telepsychology is the notion of telepresence, which was born from anecdotal references regarding the feeling of being moved from a local control room to a remote area when using a teleoperator [17]. In telepsychology, the feeling of telepresence is about the quality of the relationship between the patient and the psychologist [18]. It is an important concept to understand relational processes in telepsychology and it consists of forgetting to some extent that we are at a distance, i.e., feeling together [19]. This feeling of telepresence also seems to predict the therapeutic alliance [9]. In cognitive behavioural therapy using video conferencing, the therapeutic alliance was, for example, found to be positively correlated with the level of telepresence felt by patients [9,20]. Similarly, the evaluation of intersubjective qualities of online interactions also appears to be related to the feeling of telepresence [21]. Several research studies have demonstrated a link between the positive emotions felt by patients in CBT videoconference psychotherapy (VCP) and the level of telepresence [18].

A commonly used tool to assess telepresence is the Videoconference Telepresence Scale (VTS), which was proposed in 2006 and revised in 2018 [22]. The scale consists of seven items, which form three subscales: (1) physical presence (whether the user felt actually present in the consultation with the other person), (2) absorption (forgetting that the psychotherapist and the patient are not in the same room), and (3) social presence (feeling that the interlocutor reacts to your presence and having the impression of actively participating in exchanges with the other person) [22]. This study focuses on social telepresence in particular.

Telepresence appears to be a cornerstone of a successful telepractice for both patient and professional [9,20]. Nevertheless, in the context of the COVID-19 pandemic, little is known about the clinical and sociodemographic factors that may have influenced the effectiveness of remote consultations on the patient side [23] or about the factors that affect professionals’ perceptions of telepsychology [24]. Recent research, however, not only showed that telepsychology has the potential to improve access to care, but also that providers’ attitudes toward this innovation play a crucial role in its adoption. Some studies found that providers had an overall positive attitude toward telepsychology, even despite multiple drawbacks [25]. Additionally, there is a relationship between access to technology, experience and practitioner training on the one hand and acceptability of videoconferencing use on the other hand [24]. Studies have also shown that psychologists’ attitudes about telepsychology and subjective norms are associated with the intention to use telepsychology, which in turn is related to experience of clinical work done via telepsychology [26].

Beyond the current health crisis, telepsychology is likely to remain a major component of mental healthcare, as it might help to improve access to mental healthcare and to reduce costs of service delivery [27]. It is therefore critical to determine the impact, potential benefits and disadvantages of telepsychology not only for patients [28], but also for professionals. In general healthcare, previous work has already found that professionals were often concerned about technical and clinical quality, safety, privacy, and accountability [29,30,31] when opting for tele-health. In this paper, however, we focus particularly on the mental healthcare professionals’ point of view on telepsychology. We study the impact of their personal characteristics and training and try to explore them in relation to the concept of telepresence, which has received less attention than acceptance in the literature.

We therefore aim to present results on telepsychology practices in different European countries during the initial lockdown caused by the COVID-19 pandemic, where access to mental healthcare was highly limited. A large international online survey on the use of telepsychology, in this case focused on online psychological consultations during the COVID-19 pandemic, was set up. Qualitative analyses of actual needs and concerns relating to the use of online consultations have already been published [32] as well as quantitative exploratory analyses which primarily aimed to gain insights into determinants of telepsychology (non)adoption and experience [33]. This study builds on this existing work by focusing exclusively on mental healthcare professionals who had adopted the practice of telepsychology, exploring the extent to which differences across countries can be retrieved and investigating social telepresence.

The first goal of this study is to examine determinants of mental healthcare professionals’ perception of telepsychology in Europe during the COVID-19 pandemic. The second goal is to investigate the determinants of social telepresence and the role of telepsychology perception in social telepresence. The analysis takes into account personal characteristics such as gender, motivations, prior training, and experience of mental healthcare professionals. In the proposed conceptual model, we assume that the personal characteristics and the level of training in telepsychology of mental healthcare professionals are factors that directly influence the perception of telepsychology, invariant for country. Additionally, we assume that the level of social telepresence experienced by professionals is directly influenced by their motivation for telepsychology, personal characteristics, and level of training in telepsychology. As a result, we also presume indirect effects of training on mental healthcare professionals’ perception of telepsychology.

## 2. Materials and Methods

### 2.1. Sample

In the context of the COVID-19 pandemic, the EFPA Project Group on e-Health launched a European survey from 18 March to 5 May 2020 on the online practices of mental healthcare professionals. It was conducted through the mailing lists and social media of the European Federation of Psychologists’ Associations (EFPA) as well as national psychologists’ associations and project collaborators from different countries.

The online survey was designed to assess the extent to which mental health professionals were implementing telepsychology during the COVID-19 pandemic, their experience with this treatment modality and their concerns. Telepsychology was operationalised as online consultations, i.e., digital contact (text, audio, and/or video) for psychological counselling or psychotherapy. The survey was translated into 17 languages by local researchers and professionals in the field of psychology. This study has analysed the perception of social telepresence for mental healthcare professionals in Austria, Belgium, Denmark, Finland, France, Germany, Italy, the Netherlands, Norway, Portugal, Spain, Sweden and the UK, and relied on a subsample of participants who had opted for telepsychology.

### 2.2. Measures

*Dependent variables of interest.* The first variable was ‘Perception of telepsychology’, which was measured using the question “How comfortable do you consider yourself with (the concept of) online consultations”. It was scored on a 5-point Likert scale ranging from 1, ‘*Highly uncomfortable*’, to 5, ‘*Highly comfortable*’, with higher scores reflecting greater satisfaction and less concern.

The second variable was ‘Social telepresence’, the feeling of being connected with one another linked to the evaluation of social presence in the VTS. It was measured using the question: “*If you have started online consultations recently, how would you rate your level of telepresence (feeling of being connected with one another) during consultations?*”. It was scored on a 5-point Likert scale, ranging from 1, ‘*Very low*’, to 5, ‘*Very high*’. Higher scores reflected greater satisfaction and less concern about one’s social telepresence during telepsychology.

*Independent variables of interest.* The first set of variables of interest were ‘Personal characteristics of mental healthcare professionals’. These included age (five dummy variables with less than 35 years old as the referent category, compared to 35–44, 45–54, 55–60, over 60), gender (0 = male, 1 = female), professional status (five dummy variables with self-employed as the referent category, compared to group practice, health care organization, mental health care organization, other), and professional seniority (years of professional activity as mental healthcare professionals).

The second variable was ‘Reasons for telepractice implementation’. There were several possible motivations for mental healthcare professionals to make use of telepsychology. They included: (1) considering it a necessity from a public health point of view; (2) meeting clients’ demands; (3) not wanting to lose income; (4) wanting to provide access to care; and (5) being open-minded concerning digital mental health. As for social telepresence, the data only provide this information for mental healthcare professionals who have only newly used telepsychology.

The third set of variables was ‘Training’. Telepractice before the pandemic was captured by the following question: “*I have experience with online consultations, prior to the COVID-19 outbreak.*” (0 = no, 1 = yes). Specific training was defined as “*Have you had any specific training concerning online consultations?*” (0 = no, 1 = yes). We ran econometric models to determine the factors that drove this perception. Descriptive data for the sample are presented in Table 1 and in Appendix A.

### 2.3. Empirical Models

The following models were used:

$Y_{ij}^{1}$: Perception of the concept of telepsychology practice in Europe
$$Y_{ij}^{1}=a{\left(Training\right)}_{ij}+b{\left(Personal\;information\;on\;psychologists\right)}_{ij}+\beta_{j}+\varepsilon_{ij}^{1}$$

With *i* the individual and *j* the country

Personal characteristics of mental healthcare professionals: age group, gender, self-employed, group practice, healthcare organisation, mental healthcare organisation, years of experience; training including telepractice before the pandemic and specific training; $\beta_{j}$: the country fixed effects; $\varepsilon_{ij}^{1}$: the error terms

$Y_{ij}^{2}$: The social telepresence for new telepsychology users in Europe


$$\begin{array}{c} Y_{ij}^{2}\\ =a{\left(\mathrm{Training}\right)}_{\mathrm{ij}}+b{\left(\mathrm{Personal}\mathrm{infomation}\mathrm{on}\mathrm{psychologists}\right)}_{\mathrm{ij}}\\ +c{\left(\mathrm{Motivation}\mathrm{for}\mathrm{telepractice}\mathrm{for}\mathrm{fresh}\mathrm{psychologists}\right)}_{\mathrm{ij}}\\ +d{\left(\mathrm{Perception}\mathrm{with}\mathrm{the}\mathrm{concept}\mathrm{of}\mathrm{telepsychology}\mathrm{practices}\mathrm{in}\mathrm{Europe}\right)}_{\mathrm{ij}}+\beta_{j}+\varepsilon_{ij}^{2} \end{array}$$


With *i* the individual and *j* the country

Personal characteristics of mental healthcare professionals: age group, gender, self-employed, group practice, healthcare organisation, mental healthcare organisation, years of experience. training, including telepractice before the pandemic and specific training; motivation for telepractice, including perceived public health need, client’s demand, not losing income, guaranteeing access, technological interest; $\beta_{j}$: the country fixed effects. $\varepsilon_{ij}^{2}$: the error terms.

We used an incremental model, which is a form of the hierarchical regression model (in which variables are added or removed from a model in multiple steps), as used, for example, by Clark and Milcent [34]. This model sought to examine the explanatory power of variables that are deemed essential, together with other groups of variables. The interest of this model, compared to the stepwise model, is not to remove non-significant variables, but to test the influence of correlations of internals to one group of variables on variables in a second group. We also used a fixed effect model for nested data: here, mental healthcare professionals belong to one country. So, we have the mental healthcare professional dimension and the country dimension. The nested data here consist of two levels of sampling, with observations sampled at Level One (mental healthcare professionals) nested (or clustered) within units sampled at Level Two (country). Hierarchical linear modelling is used here with fixed effects.

We investigated the determinants that impacted mental healthcare professionals’ telepsychology experience, i.e., their perception of the concept of telepsychology practices, as well as the factors that positively impacted “teleperception”. The database is a nested panel of data at two levels: respondent and country. The data also showed an unbalanced structure at two levels: the number of respondents per country differs from country to country. We wanted to work at the individual level in order to take advantage of the variability at the individual level. We therefore used a fixed effect model to control for country specificities and to provide a respondent-level analysis: thus, the effect of mental healthcare professionals’ specificities on the perception of telepsychology and social telepresence of mental healthcare professionals was identified only by seeing how mental healthcare professionals’ specificities changed within a given country. No cross-country information was used in the estimation of these coefficients.

This approach conditioned the sum of an individual’s feelings over a country. The fixed effects approach [35,36,37,38,39] controlled for any interaction between the mental healthcare professionals surveyed and the country variables. Each fixed effect probit model was studied through a multiple regression using STATA SE version 16.

## 3. Results: Perception of Telepsychology

### 3.1. Descriptive Data

Table 1 reports the descriptive data for the dependent variables in the study, with more details to be found in Appendix A. The dataset was unbalanced, as the number of surveyed mental healthcare professionals varied for each country.

Spain, Portugal, Denmark, and the UK were the countries where mental healthcare professionals felt most comfortable with the notion of telepsychology. On the other hand, Finland, Belgium, Austria, and the Netherlands were the countries where mental healthcare professionals were least comfortable with telepsychology. Higher levels were reported in Germany, Denmark, Austria and the UK. Participants from these countries reported experiencing a higher level of social telepresence during telepsychology, compared to lower levels in Finland, Sweden, the Netherlands and Belgium.

There were clear disparities between countries with regard to the correlations between both concepts. Two atypical situations were Austria (almost zero correlation) and Spain (non-significant correlation). The correlation then oscillated from *r* = 0.18 (Norway) to *r* = 0.53 (United Kingdom). There did not appear to be a relationship between the level of correlation and high or low scores in the perception of telepsychology or social telepresence. We controlled for cross-countries disparities using fixed effect models. The interpretation of the independent variables was then completed as if mental healthcare professionals were similar, regardless the country. This econometric model was used as we had nested panel data at the level of mental healthcare professionals and countries. It allowed to control for country-invariant attributes of participants.

### 3.2. Do Mental Healthcare Professionals’ Characteristics Play into the Perception of Telepsychology?

In this sample, French mental healthcare professionals were mostly women (90%), as was the case in Denmark, Belgium, the Netherlands, and Portugal. In Germany and Spain, the context was quite different with 64% and 68%, respectively, of female mental healthcare professionals. The sample was younger in Belgium, France, and Portugal (41 years old on average), compared to other European countries (49 years old in Austria, Denmark and the UK). In line with this, they had fewer years of professional seniority in Belgium and France (13 years) than what was observed in the samples of other European countries (21 years in Austria and 17 years in Denmark, the UK, and Spain). There was substantial heterogeneity regarding the context of practice across countries. In Italy, 78% of mental healthcare professionals were self-employed, whereas in The Netherlands and Sweden, less than 20% were. In Norway or Finland, one third worked in a health organization (Appendix A). To control for this variability in mental healthcare professionals’ practices across countries, we mobilized a fixed effect country model to explore perceptions of telepsychology.

In many applications, including econometrics and biostatistics, a fixed effects model refers to a regression model in which the group means are fixed (non-random) as opposed to a random effects model in which the group means are a random sample from a population. This model captured country specificities in a country-dummy variable. We analysed mental healthcare professionals’ perception as if the countries were comparable in terms of specificities. Using a stepwise multiple regression, we first considered only demographic variables. Then, we added practice conditions.

In Table 2 and Table 3, a country dummy, a set of variables specific to country that are invariant over time, captured the country fixed effect. There were as many country dummies as countries (and consequently no intercept because of the strict collinearity). Results were calculated from the model based on the independent variables of interest. “Reference” is the reference group used for comparison.

Table 2 Columns (1), (2), and (3) refer to incremental models 1 to 3. Model 1 refers to the Demographic variables included in the personal characteristics of mental healthcare professionals. Model 2 refers to Personal characteristics of mental healthcare professionals. Model 3 refers to Personal characteristics of mental healthcare professionals + Training. Table 2, Column (1) shows that female mental healthcare professionals had a more negative perception of telepsychology. Even after controlling for professional status and professional seniority (in years), a similar result was obtained (Table 2, Column (2)). 

Similarly, after adding control for training (telepsychology practice prior to the pandemic and any specific training for telepsychology), the results showed a negative effect of gender on the perception of telepsychology (Table 2, Column (3)). Additionally, Table 2 shows that the seniority of the mental healthcare professionals compared to the youngest (under 35 years old), improved the perception of telepsychology. However, this effect was captured by the effect of training on the perception of telepsychology (Table 2, Column (3)). Working in a healthcare organisation, rather than being self-employed, made mental healthcare professionals less positive about telepsychology (Table 2, Columns (1) & (2)). However, this effect was captured by the effect of training on the perception of telepsychology (Table 2, Column (3)). The addition of training as a variable removed the effect of work status.

Taken together, the main results suggested that female mental healthcare professionals had a less positive perception of telepsychology and that seniority in the profession and the fact of working independently were related to a more positive perception of telepsychology.

### 3.3. Does Training Have an Impact on the Perception of Telepsychology?

We studied the effect of training—operationalised as telepsychology practice before the pandemic as well as specific training—with personal characteristics of mental healthcare professionals as a control.

First, in terms of percentages, some figures on mental healthcare professionals who practiced telepsychology before the pandemic. Prior to the COVID-19 outbreak, the percentage of professionals with experience in telepsychology was 58% in Spain and 54% in the UK, respectively. In contrast, less than a quarter of mental healthcare professionals practiced online in Belgium and France. Regarding specific training for telepsychology, very few European countries reported specific training among mental healthcare professionals. In Austria, Finland, and the UK, more than one fifth of respondents had specific training in support of telepsychology. In France, it dropped to less than one in twenty and 6.6% in Belgium (Appendix A). We used an econometric model to control for this country heterogeneity: a country fixed effect model.

Controlling for mental healthcare professionals’ personal characteristics, training had a very positive impact on the perception of telepsychology.

The training dimension was captured by two proxies in Table 2: (1) Telepractice before the pandemic and (2) Specific training. As shown Table 2, for these two proxies, the coefficients were significant. Then we were able to conclude that not only previous experience (in years) of telepsychology practices impacted the perception of telepsychology positively; but also that specific training in itself improved the perception of telepsychology. This result was found after controlling for the mental healthcare professionals’ characteristics. As a sensitivity analysis, we ran the model without any additional control variables. The results were similar.

Results suggested that training—telepsychology practice before the pandemic and specific training—had a positive impact on the perception of telepsychology.

### 3.4. Do Mental Healthcare Professionals’ Personal Characteristics Have an Impact on the Perception of Social Telepresence?

Table 1 shows the number of mental healthcare professionals surveyed by country. As global features, we observed disparities between countries in the level of social telepresence of the mental healthcare professionals, as presented in Table 1. The average was higher in Germany (*M* = 4.00) and Denmark (*M* = 3.99), as compared to in Finland (*M* = 3.47) and Sweden (*M* = 3.55). The econometric model took these disparities between countries into account.

Table 3 presents the results. Table 3 Columns (1), (2), (3), (4), and (5) refer to incremental models 1 to 5. Model 1 refers to the demographic variables included in the personal characteristics of mental healthcare professionals. Model 2 refers to Personal characteristics of mental healthcare professionals. Model 3 refers to Personal characteristics of mental healthcare professionals + Training. Model 4 refers to Personal characteristics of mental healthcare professionals + Training + Motivations to start telepsychology. Model 5 refers to Personal characteristics of mental healthcare professionals + Training + Motivations to start telepsychology + Perception.

Using a stepwise multiple regression, we showed the relationship between characteristics of professional status, professional seniority (in years), and social telepresence level. The demographic characteristics of mental healthcare professionals—age and gender—played into social telepresence (Table 3, column (1)). However, by adding professional status and professional seniority, we found that only gender affects social telepresence: being a female psychologist positively affected the level of social telepresence (Table 3, column (2)). When we added training, this result remained, as well as when we added reasons to practice telepsychology (Table 3, columns (3) and (4)). Adding perception of telepsychology did not change this result (Table 3, column (5)).

Employment status, however, did affect social telepresence. Compared to self-employed mental healthcare professionals, working in a healthcare organisation reduced the feeling of social telepresence. This result was obtained regardless of the set of variables controlled (Table 3, columns (2)–(5)). Mental healthcare professionals’ year of professional seniority positively affected feeling of social telepresence (Table 3, columns (2)–(5)).

Results therefore suggested that mental healthcare professionals’ characteristics had an impact on the feeling of social telepresence: what had a positive influence was being a female mental healthcare professional, professional seniority as a mental healthcare professional and being self-employed compared to working in a healthcare organisation.

### 3.5. Does Training Impact Social Telepresence?

The training dimension was captured by two proxies in Table 3. These two variables, ‘Telepsychology practices before the pandemic’ and ‘Specific training’, were significant. The variable present when social telepresence was very high, ‘Telepsychology practices before the pandemic’ had a coefficient of 0.033 and ‘Specific training’ had a coefficient of 0.021. We therefore concluded that training had a positive impact on the feeling of social telepresence, even when taking the mental healthcare professionals’ personal characteristics into consideration. This could be due to telepsychology experience prior to the pandemic or specific training to prepare for telepsychology. This result remained when controlling for motivations to practice telepsychology. It also held when adding the level of perception of telepsychology in the set of explanatory variables.

Results therefore suggested that the accumulation of different forms of training promoted a sense of social telepresence.

### 3.6. Do Personal Motivations Have an Impact on Social Telepresence?

Looking at the reasons for starting telepsychology, preliminary analyses suggested heterogeneity in mental healthcare professionals’ responses across European countries. The public health reason was a major factor for the UK, Belgium, Denmark, and Finland (around 75%). Another major reason was access to healthcare (56%), except for France where the percentage was lower (40%) (See Appendix A) [39]. Reasons such as ‘Client’s demand’ or ‘Not losing income’ were quoted by a quarter of respondents. Austria, France, and Portugal stand out for the motivation of being open-minded to telepsychology. The model used—a country fixed effect model—allowed us to control for this country heterogeneity. We obtained results as if countries were similar, with the country-dummy variable capturing the national specificities.

In Table 3, Column (4), we focused on reasons for starting telepsychology during the lockdown, considering the public health motivation, client demand, income motivation, healthcare access, and telepsychology openness. In this model, we controlled for mental healthcare professionals’ characteristics, and personal training.

We found that some motivations had a positive impact on the social telepresence of the mental healthcare professionals. These motivations were public health, client demand, and access to healthcare (Appendix A). To be open-minded to telepsychology played a role in social telepresence, but this factor was correlated with the perception of telepsychology to explain social telepresence. In other words, when controlling for perception of telepsychology, the open-mindedness to telepsychology reason was not significant anymore (Appendix A). Income-related motivation negatively impacted mental healthcare professionals’ feeling of social telepresence.

Results then suggested that income-related motivation worsened the feeling of social telepresence and that being open-minded to telepsychology was correlated with the perception of telepsychology to explain social telepresence.

### 3.7. Does the Perception of Telepsychology Impact Social Telepresence?

Table 3, Column (5) shows that in addition to the other factors that foster social telepresence, a positive perception of telepsychology had a positive impact on the feeling of social telepresence.

Results suggested that the perception of telepsychology had a positive impact on social telepresence.

### 3.8. Sensitivity Analysis

The sample used was an unbalanced dataset. In the sample, the number of observations was much higher in the Italian subsample than for the other countries (Appendix A). We dealt with this using fixed effect models. However, the results may remain debatable. As a sensitivity analysis, we excluded the Italian mental healthcare professionals surveyed from the sample, which allowed us to see how the Italian information affected the average results.

Some results on the mental healthcare professionals’ characteristics were not consistent with the results previously found on the whole sample. However, we found that the perception of telepsychology depended significantly and positively on training (telepsychology practices before the pandemic and specific training), as it was found in the whole sample. In the sub-sample of new mental healthcare professionals as users of telepsychology, training and positive perception of telepsychology were important factors to foster social telepresence. This result was in line with what we found in the overall sample. Therefore, we concluded that with or without the Italian subsample, the main results remained.

## 4. Discussion and Conclusions

In this paper, we studied the determinants of the perception of telepsychology in a large cross-sectional survey study. Subsequently, we analysed the drivers of social telepresence. We observed disparities between countries (Appendix A) and used a fixed effects econometric model to accommodate these. This study, using a fixed effects approach in a large sample of mental healthcare professionals (allowing to eliminate interactions between the surveyed psychologists that were invariant across countries), is an addition to the study of telepsychology in Europe, where many studies on telepsychology focus on clients’ perceptions.

Results suggested that female mental healthcare professionals were more likely to have a negative perception of telepsychology, all else being equal. Training—consisting of both telepsychology practice before the pandemic and specific training—was also a key factor impacting perception of telepsychology [40].

The second interest is social telepresence. The feeling of social telepresence appeared to depend on the mental healthcare professionals’ personal characteristics. Female mental healthcare professionals had a higher feeling of social telepresence, even after controlling for all other explanatory variables. The reasons for this remain to be explored. Despite the general positive association between perception and social telepresence, women generally tended to have a more negative perception of telepsychology but a higher social telepresence. Could it be that women are more sensitive than men to relational aspects, which would leave them to have a priori reservations about telepsychology while also predisposing them to being more sensitive to the effects of social telepresence? Previous work has already hinted at gender differences in therapeutic relationships, with female therapists being positioned as more emotionally supportive while male therapist could be seen as more goal-oriented [41,42]. Professional situation was also related to social telepresence. According to the results, social telepresence was higher in those self-employed (and therefore probably freer in their choice of practice) and those with professional seniority as mental healthcare professionals (which may give more confidence, and therefore adaptability to changes in this context). Training, in whatever form (personal experience or specific training), also seemed to be a way to foster social telepresence. Finally, motivations for telepsychology adoption and acceptance were also related to social telepresence. Interestingly, financial motivations worsened social telepresence while openness to telepsychology was correlated with a positive perception of telepsychology to explain social telepresence.

These study results ultimately highlighted the role of “telepsychology perception” as a driver of social telepresence. The results also showed a correlation between telepsychology training and a good perception of telepsychology. Findings suggested that appropriate telepsychology training should enhance social telepresence directly and through experience of telepsychology in practice. This factor could therefore improve the acceptance of telepsychology among professionals

This study also has some limitations that require discussion. Since it is a cross-sectional study, findings need to be confirmed in longitudinal and experimental designs. As we aimed for brevity of the entire questionnaire (at that moment in time when mental healthcare professionals were under exceptional pressure due to the pandemic), only one item was used to evaluate an aspect of telepresence. No telepresence scale was included in our questionnaire, so we only evaluated the dimension of social telepresence.

The pandemic was a time when some mental healthcare professionals began an online practice under difficult conditions. Some of them had no prior experience or training in telepsychology. This raises questions about ensuring a clinical and ethical framework under these conditions and improving these critical terms in the future. According to the APA Guidelines for Telepsychology: “Psychologists who provide telepsychology services strive to take reasonable steps to ensure their competence with both the technologies used and the potential impact of the technologies on clients/patients”. These are certainly avenues to be explored to strengthen the framework for telepsychology in Europe, including a reflection on adaptations of the training framework to the practice of psychologists. We note that rare initiatives are beginning to appear in Europe, such as the Erasmus Mundus CYBER Master (https://www.cyber-t.eu, accessed on 25 September 2022), which opened its doors in 2022.

In the present context, and given the increase of telepsychology, issues related to the terminology, methodology and ethics of telepsychology are essential. Our results help to promote reflection among professional federations such as EFPA or FFPP (https://ffpp.net/de-la-cyberpsychologie-a-la-teleconsultation, accessed on 25 September 2022) and to build a solid telepsychology framework to support mental healthcare professionals and foster their acceptance of telepsychology.

## Figures and Tables

**Table 1 jcm-12-02147-t001:** Perception of telepsychology and social telepresence as reported by European mental healthcare professionals, and the correlation between these concepts.

	Perception of Telepsychology			Social Telepresence			
Country	N	*M*	*SD*	N	*M*	*SD*	*r*
Austria	65	3.84	0.89	43	3.97	1.12	0.01
Belgium	556	3.83	0.81	404	3.56	0.88	0.51 **
Denmark	76	4.06	0.79	68	3.99	0.76	0.41 **
Finland	141	3.59	1.03	79	3.47	1.04	0.48 **
France	491	3.94	0.70	222	3.84	0.81	0.41 **
Germany	168	3.97	0.76	92	4.00	0.84	0.23 *
Italy	4604	3.95	0.75	3605	3.75	0.81	0.43 **
Netherlands	81	3.87	0.94	68	3.56	0.97	0.36 **
Norway	258	4.02	0.87	189	3.83	0.76	0.19 **
Portugal	267	4.19	0.69	165	3.86	0.76	0.39 **
Spain	31	4.41	0.50	21	3.85	0.97	0.30
Sweden	395	3.93	0.96	223	3.55	1.00	0.45 **
UK	50	4.03	0.81	28	3.89	0.69	0.54 **

Note. * *p* < 0.05, ** *p* < 0.01.

**Table 2 jcm-12-02147-t002:** Econometric model predicting the perception of telepsychology among European mental healthcare professionals.

	Perception of Telepsychology		
	(1)	(2)	(3)
Mental healthcare professionals’ personal characteristics			
<35 years	Reference	Reference	Reference

>35–40 years	0.09 *	0.06	−0.01
	(0.039)	(0.041)	(0.039)
>40–50 years	0.25 **	0.20 **	0.14 **
	(0.043)	(0.052)	(0.050)
>50–55 years	0.13 *	0.14	0.11
	(0.063)	(0.076)	(0.073)
>55 years	0.18 **	0.11	0.10
	(0.056)	(0.086)	(0.081)
Gender: female	−0.25 **	−0.25 **	−0.16 **
	(0.042)	(0.040)	(0.038)
Self-employed		Reference	Reference
Group practice		0.15	0.20 *
		(0.103)	(0.098)
HC organisation		−0.11 *	−0.01
		(0.047)	(0.045)
MHC organisation		−0.12 **	−0.05
		(0.044)	(0.042)
Years of experience		0.00	−0.00
		(0.002)	(0.002)
Training			
Telepsychology practice before the pandemic			0.35 **
			(0.047)
Specific training			0.77 **
			(0.029)
Country Fixed Effects	Yes	Yes	Yes
R-Squared	0.025	0.013	0.109

Notes: *, **: Student’s *t*-test; * *p* < 0.05, ** *p* < 0.01; Standard errors in parentheses. Reference: reference group used for comparisons.

**Table 3 jcm-12-02147-t003:** Econometric model predicting social telepresence in telepsychology and the characteristics of new European mental health professional users.

	Social Telepresence of European Mental Healthcare Professionals New to Telepsychology (Level)				
	(1)	(2)	(3)	(4)	(5)
Personal characteristics of mental healthcare professionals					
<35 years	Reference	Reference	Reference	Reference	Reference
>35–40 years	0.06	0.01	−0.01	0.01	0.01
	(0.031)	(0.032)	(0.032)	(0.032)	(0.029)
>40–50 years	0.15 **	0.03	0.01	0.03	−0.01
	(0.034)	(0.041)	(0.041)	(0.041)	(0.037)
>50–55 years	0.18 **	0.07	0.05	0.07	0.03
	(0.049)	(0.061)	(0.060)	(0.060)	(0.055)
>55 years	0.20 **	−0.00	−0.02	0.00	−0.04
	(0.044)	(0.069)	(0.068)	(0.067)	(0.062)
Gender: female	0.10 **	0.08 **	0.11 **	0.12 **	0.15 **
	(0.032)	(0.031)	(0.030)	(0.030)	(0.028)
Self-employed	Reference	Reference	Reference	Reference	Reference
Group practice		−0.05	−0.04	−0.07	−0.11
		(0.078)	(0.077)	(0.076)	(0.070)
HC organisation		−0.15 **	−0.15 **	−0.18 **	−0.19 **
		(0.041)	(0.041)	(0.041)	(0.038)
MHC organisation		−0.18 **	−0.18 **	−0.20 **	−0.20 **
		(0.037)	(0.036)	(0.036)	(0.033)
Professional seniority: Years of activity		0.01 **	0.01 **	0.01 **	0.01 **
		(0.002)	(0.002)	(0.002)	(0.002)
Training					
Telepsychology practice before the pandemic			0.18 **	0.16 **	0.10 **
			(0.036)	(0.036)	(0.033)
Specific training			0.21 **	0.21 **	0.07 **
			(0.022)	(0.022)	(0.021)
Motivations for telepsychology for novel mental healthcare professionals					
Public health				0.19 **	0.12 **
				(0.025)	(0.023)
Client				0.15 **	0.10 **
				(0.026)	(0.024)
Income				−0.12 **	−0.11 **
				(0.028)	(0.026)
Access				0.10 **	0.09 **
				(0.023)	(0.021)
Techno				0.06 *	0.01
				(0.029)	(0.027)
Perception					0.28 **
					(0.009)
Country Fixed Effects	Yes	Yes	Yes	Yes	Yes
R-squared	0.022	0.017	0.037	0.059	0.206

Notes: *, **: Student’s *t*-test; * *p* < 0.05, ** *p* < 0.01; Standard errors in parentheses. Reference: reference group used for comparisons.

## Data Availability

The data presented in this study are available on request from the corresponding author.

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
