# Peer review of "Telepsychology in Europe since COVID-19: How to Foster Social Telepresence?"

_jcm, 2023, doi:10.3390/jcm12062147_

Round 1

Reviewer 1 Report

This study focuses on an important question salient in the teletherapy research and includes a diverse international dataset, which are great strength of the study. However, it also has many points to address, namely the following: 

Abstract

The definition of telepresence is vague, there are very precise definitions (i.e.., sense that being in the same room to use. The findings in the abstract are a little unclear, maybe because of language limitations. Implications/discussion of the study is missing from the abstract.

Introduction: The definition of telepresence is missing. Please support why you state it is a "cornerstone of a successful tele practice for patient and professional". Moreover, it appears the paper is about teletherapy and not telepsychology in general. The provided telepsyhology definition is a very broad one provided by APA in 2013, much has happened in the field of teletherapy since then, especially during the pandemic, in terms of research and definition of terms and concepts.

The authors state that "A lot of research in telepsychology has focused on patients' experiences, but very few studies so far have targeted psychologists" - I believe there is more research now on therapists' views and attitudes on teletherapy then that of patients', especially given the growing number of studies published during the pandemic.

The proposed conceptual model presented at the end of the first paragraph is not explained or introduced, neither is the study itself which is also presented within the first paragraph.

The authors state that "According to Patel et al. [7], the COVID-19 pandemic has been associated with a marked increase in remote mental health consultation, particularly among younger patients" - please provide context, as for example the in North America, almost all mental health services, and practically all individual therapies needed to move online at certain phases of the pandemic, not simply an increase in remote mental health interventions was observed, and not only among a particular population. This was somewhat different in Europe and providing some explanation and context would be helpful, for example what determined if practices needed to more online, who made the decision etc. This may also impact the study results, in case transitioning to online was based on individual decision by the therapist, and thus pose a selection bias for the analyses in the study (i.e., including those who provided some online consultation).

Please define telepsychiatry, especially that it seems to be used interchangeably with telepsychology.

The authors state "Furthermore, almost nothing is known relating to the professionals", that is, demographic and professional factors that are associated with their experiences and views on telepsychology - Actually, there is quite a few studies now published on these factors. Moreover, while it seems the authors mean teletherapy throughout the paper, they use the term "telepsychology", please clarify. Later they also use the term "teleconsultation".

For a review study on concerns about telepsychology I suggest to include Connolly et al 2020 study.

Please define telepresence when introduced.

Methods

Sample: first 2 sentences are redundant, they were already described. It would be helpful to see the overall sample size and range by country here.

"The current study used a subsample of participants who provided answers to be used for a quantitative analysis " - this sounds like the original study was not quantitative, just this subsample, can you clarify?

"Perfect transcription of questionnaire" - I believe you meant "translation"

Why were participants excluded from the analysis if they used another language than their national language(s) or English? How did the authors know what was the first language of the participant? This is unusual process.

Perception of online consultations was assessed by a single question. Telepresence was defined by "feeling of being connected with one another" which does not match the commonly used definition of telepresence (see for example Bouchard et al, 2018) bar rather some aspect of a relational or alliance process. This is misleading as the study item in fact does not ask about telepresence. This was also measured by a single item.

Results

The descriptive data in Table 1 does not describe the sample but the means and correlations between variables, therefore information about the sample demographics (age, gender, professional background etc.) is needed.

The results provide country comparison data that was not part of the research question.

The authors phrase associations between variables by referring the impact of variable on the other, since we cannot know about causation in this cross sectional study, I suggest to refrain from such indications in general, but especially in the case of the last statement: "The results then suggest that the perception of teleconsultation has a positive impact on telepresence." 

Discussion

The authors conclude that perception of teleconsultation is a driver of telepresence, which is not supported by the data. We know that there is a relationship between the two items (not questionnaires), and that telepresence refers more to the relationship than telepresence as understood in the previous literature, therefore this connection can be only stated very tentatively if at all.

Author Response

Point 1: Abstract: The definition of telepresence is vague, there are very precise definitions (i.e.., sense that being in the same room to use. The findings in the abstract are a little unclear, maybe because of language limitations. Implications/discussion of the study is missing from the abstract.

Response 1: We thank you for your comment, we added a definition of telepresence in the abstract: which consists of forgetting to some extent that we are at a distance, in feeling together (Haddouk, Bouchard, 2022).

We also mentioned in the abstract that we focused on social telepresence in the present paper.

We added information on the model and how it addresses the problem.

Following your comments, we have also modified the title of the article and we have removed the word: "teleconsultation" from the keywords.

Point 2: Introduction: The definition of telepresence is missing. Please support why you state it is a "cornerstone of a successful tele practice for patient and professional".

Response 2: We thank you for this point and we have added elements of the definition of telepresence in videoconferencing telepsychotherapy.

We also added some references about the feeling of telepresence and its correlation with positive emotions felt by patients in CBT videoconference psychotherapy (VCP) and the therapeutic alliance. We also mentioned tools to evaluate telepresence, such as the Videoconference Telepresence Scale (VTS) (Bouchard and Robillard, 2006, 2018).

Point 3: Moreover, it appears the paper is about teletherapy and not telepsychology in general. The provided telepsychology definition is a very broad one provided by APA in 2013, much has happened in the field of teletherapy since then, especially during the pandemic, in terms of research and definition of terms and concepts.

Response 3: We thank you for your remark and totally agree on this point. To be clearer on our purpose, we have removed the word: "teleconsultation" from the keywords. We have also been careful to avoid confusion between the terms: "teleconsultation" and: "telepsychology" in the article. We have used the term: "telepsychology" to qualify the results we have obtained from psychologists. The term: "teleconsultation" appears only in reference to the remote consultations proposed by psychiatrists, in the framework of a medical activity.

Point 4: The authors state that "A lot of research in telepsychology has focused on patients' experiences, but very few studies so far have targeted psychologists" - I believe there is more research now on therapists' views and attitudes on teletherapy then that of patients', especially given the growing number of studies published during the pandemic.

Response 4: We thank you for this point and we added some references about therapists’ views and attitudes on telepsychology, such as:

  • Parisi, K.E. ; Dopp, A.R. & Quetsch, L.B. Practitioner use of and attitudes towards videoconferencing for the delivery of evidence-based telemental health interventions: A mixed methods study. Internet Interventions, 2021, 26 : 100470
  • Connolly, S. L. ; Miller, C. J. ; Lindsay, J. A. & Bauer, M. S. A systematic review of providers' attitudes toward telemental health via videoconferencing. Clinical Psychology: Science and Practice, 2020, 27(2), e12311. 10.1111/cpsp.12311
  • McKee, G.B., Pierce, B.S., Donovan, E.K., & Perrin, P.B. Examining models of psychologists' telepsychology use during the COVID‐19 pandemic: A national cross‐sectional study. Clinical Psychology, 2021, (77(10): 2405-2423.
  • Zentner, K.; Gaine, G.; Ethridge, P.; Surood, S. & Abba-Aji, A. Clinicians’ Attitudes Toward Telepsychology in Addiction and Mental Health Services, and Prediction of Postpandemic Telepsychology Uptake: Cross-sectional Study. 2022. JMIR Publications, 6(5) :e35535, doi: 10.2196/35535
  •  

Point 5: The proposed conceptual model presented at the end of the first paragraph is not explained or introduced, neither is the study itself which is also presented within the first paragraph.

Response 5: We thank you for this comment and we have modified our conceptual model (Figure 1) and we included a better presentation in the introduction. We focused on the variables that we tested in our analysis: social telepresence and the influence of perception of telepsychology, motivation for telepsychology, training and psychologists’ characteristics. We assumed this model invariant for country and we added it to the model’s presentation in the introduction. We tested our theoretical model and we presented results related to it in part 3. (Results).

Point 6: The authors state that "According to Patel et al. [7], the COVID-19 pandemic has been associated with a marked increase in remote mental health consultation, particularly among younger patients" - please provide context, as for example the in North America, almost all mental health services, and practically all individual therapies needed to move online at certain phases of the pandemic, not simply an increase in remote mental health interventions was observed, and not only among a particular population. This was somewhat different in Europe and providing some explanation and context would be helpful, for example what determined if practices needed to more online, who made the decision etc. This may also impact the study results, in case transitioning to online was based on individual decision by the therapist, and thus pose a selection bias for the analyses in the study (i.e., including those who provided some online consultation).

Response 6: We thank you for this remark, and we chose to take off this part of the sentence, because we don’t develop it afterwards. We agree on the fact that the Covid-19 pandemic has led to an increase in publications. But if there is an increase in publications concerning professionals, many of this researchis American: it is interesting, with our article, to look elsewhere, for example on the European side. Whilef in Europe we are dealing with westernized countries and comparable levels of training among psychologists, there is a great cultural diversity in the familiarization with new technologies applied to the field of mental health

When we consider the situation in the countries of Europe, we are dealing not only with very diverse cultural contexts relating to the relationship between professionals and new technologies, but also with equally diverse contexts of patient care. It is important to keep this in mind, but it is equally difficult to report on it in detail in an article. This does not invalidate the value of a comparative empirical approach. Designed in a specific context with an expectation of a quick response, this type of approach confronts complex design, interpretation, and translation issues that pose unquestionable methodological problems.

The increase and diversification of research on the attitudes of mental health professionals and the acceptance of this modality justifies our contribution, but in such a vast field we have focused on specific aspects that participate in a set:

(a) online consultations, within the more general framework of telepsychology ;

(b) the role that telepresence can play in the formation of these attitudes and experiences, since it has received little attention.

Point 7: Please define telepsychiatry, especially that it seems to be used interchangeably with telepsychology.

Response 7: We thank you for your remark and we agree on this point. We added a definition of telepsychiatry in the introduction.

Point 8: The authors state "Furthermore, almost nothing is known relating to the professionals", that is, demographic and professional factors that are associated with their experiences and views on telepsychology - Actually, there is quite a few studies now published on these factors. Moreover, while it seems the authors mean teletherapy throughout the paper, they use the term "telepsychology", please clarify. Later they also use the term "teleconsultation".

Response 8: We thank you for this remark, and as we mentioned in Response 3, we tried to be clearer on our purpose and we have removed the word: "teleconsultation" from the keywords. We have also been careful to avoid confusion between the terms: "teleconsultation" and: "telepsychology" in the article. We have used the term: "telepsychology" to qualify the results we have obtained from psychologists. The term: "teleconsultation" appears only in reference to the remote consultations proposed by psychiatrists, in the framework of a medical activity.

Point 9: Methods: Sample: first 2 sentences are redundant, they were already described. It would be helpful to see the overall sample size and range by country here.

Response 9: We agree and we changed these sentences in the text. We also added data about the overall sample size in Appendix 1. We thank you for this remark.

Point 10: "The current study used a subsample of participants who provided answers to be used for a quantitative analysis " - this sounds like the original study was not quantitative, just this subsample, can you clarify?

Response 10: The large European survey of the EFPA also included open-ended questions that provided insight into experienced difficulties with online consultations. An in-depth qualitative analysis of these responses has already been published (De Witte et al., 2021) so we do not go into the qualitative data in the current report.

Point 11: "Perfect transcription of questionnaire" - I believe you meant "translation"

Response 11: We agree and we thank you for this remark. We changed: “transcription” by: “The survey was translated in 17 languages” in the text.

Point 12: Why were participants excluded from the analysis if they used another language than their national language(s) or English? How did the authors know what was the first language of the participant? This is unusual process.

Response 12: We thank you for this remark and we noticed that we made a mistake. This section in the article was based on the qualitative study that we made with the survey data (Van Daele, T.; Mathiasen, K; Carlbring, P.; Bernaerts, S.; Brugnera, A.; Compare, A.; Duque, A.; Eimontas, J.; Gosar, D.; Haddouk, L.; Karekla, M.; Larsen, P.; Lo Coco, G.; Nordgreen, T.; Salgado, J.; Schwerdtfeger, A.R.; Van Assche, E.; Willems, S.; De Witte, N.A.J. Online consultations in mental healthcare: Modelling determinants of use and experience based on an international survey study at the onset of the pandemic. Internet Interv. 2022 Sep 5;30:100571), while the current study does use the full sample. For the qualitative analysis, language was a requirement to be able to include the participants (if one of the co-authors could not understand the response, we could not analyse it of course). It doesn’t really apply for this quantitative analysis since participants could just select one of the languages and complete the multiple choice questions. We therefore cut this paragraph altogether.

Point 13: Perception of online consultations was assessed by a single question. Telepresence was defined by "feeling of being connected with one another" which does not match the commonly used definition of telepresence (see for example Bouchard et al, 2018) bar rather some aspect of a relational or alliance process. This is misleading as the study item in fact does not ask about telepresence. This was also measured by a single item.

Response 13: We thank you for this comment, we agree and we added some precisions about this point in the text: “As we aimed for brevity of the entire questionnaire, only one item was used to evaluate an aspect of telepresence, which is definitely a limitation of our method. Thus, no telepresence scale was included in our questionnaire”, which is also a limitation in the interpretation of our results.

However, we note that the question we used refers to the evaluation of "social presence" in the (VTS) (Bouchard and Robillard, 2006, 2018). (feeling that the interlocutor reacts to your presence and the impression of actively participating in exchanges with the interlocutor).”

Point 14: Results: The descriptive data in Table 1 does not describe the sample but the means and correlations between variables, therefore information about the sample demographics (age, gender, professional background etc.) is needed.

Response 14: We agree on this point, we added a description of the sample in Appendix 1.

Point 15: The results provide country comparison data that was not part of the research question.

Response 15: We mentioned in the introduction : “The current study builds on this existing work by focusing exclusively on psychologists who had adopted the practice of telepsychology, exploring the extent to which differences across countries can be retrieved.” In the results part, we provide some informations about this comparison.

Point 16: The authors phrase associations between variables by referring the impact of variable on the other, since we cannot know about causation in this cross sectional study, I suggest to refrain from such indications in general, but especially in the case of the last statement: "The results then suggest that the perception of teleconsultation has a positive impact on telepresence."

Response 16: We thank you for this remark and we agree. Given that it is a cross-sectional study, we tried to be a bit more careful in our phrasing. We also added to the limitations or suggestions for further research that the associations should be confirmed in a longitudinal or experimental design.

Also, considering the limitation about the one single item and the definition of telepresence, I moderated our results and talked about : “social telepresence”.

Point 17: Discussion: The authors conclude that perception of teleconsultation is a driver of telepresence, which is not supported by the data.

Response 17: We thank you for this point, we re-phrased this part in a more tentative way and we tried to support why we think the perception of telepsychology is important.

Point 18: We know that there is a relationship between the two items (not questionnaires), and that telepresence refers more to the relationship than telepresence as understood in the previous literature, therefore this connection can be only stated very tentatively if at all.

Response 18: We thank you for this comment and we changed a bit our direction, considering only one item was used to evaluate an aspect of telepresence, which is definitely a limitation of our method, as we aimed for brevity of the entire questionnaire. We changed our results presentation, considering that we were only talking about “social telepresence”.

Response 1: We thank you for you comment, we added a definition of telepresence in the abstract This phenomenon consists in some way in forgetting that we are at a distance, in feeling together (Haddouk, Bouchard, 2022).

We also mentioned in the abstract that we focused on social telepresence in the present paper.

We added information on the model and how it allows answering the problem.

We also specify that following your comments, we have modified the title of the article and we have also removed the word: "teleconsultation" from the keywords.

Point 2: Introduction: The definition of telepresence is missing. Please support why you state it is a "cornerstone of a successful tele practice for patient and professional".

Response 2: We thank you for this point and we have added elements of definition of telepresence in videoconferencing telepsychotherapy.

We also added some references about the feeling of telepresence and its correlation with positive emotions felt by patients in CBT videoconference psychotherapy (VCP) and the therapeutic alliance. We also mentioned tools to evaluate telepresence, such as the Videoconference Telepresence Scale (VTS) (Bouchard and Robillard, 2006, 2018).

Point 3: Moreover, it appears the paper is about teletherapy and not telepsychology in general. The provided telepsychology definition is a very broad one provided by APA in 2013, much has happened in the field of teletherapy since then, especially during the pandemic, in terms of research and definition of terms and concepts.

Response 3: We thank you for your remark and totally agree on this point. To be clearer on our purpose, we have removed the word: "teleconsultation" from the keywords. We have also been careful to avoid confusion between the terms: "teleconsultation" and: "telepsychology" in the article. We have used the term: "telepsychology" to qualify the results we have obtained from psychologists. The term: "teleconsultation" appears only in reference to the remote consultations proposed by psychiatrists, in the framework of a medical activity.

Point 4: The authors state that "A lot of research in telepsychology has focused on patients' experiences, but very few studies so far have targeted psychologists" - I believe there is more research now on therapists' views and attitudes on teletherapy then that of patients', especially given the growing number of studies published during the pandemic.

Response 4: We thank you for this point and we added some references about therapists’ views and attitudes on telepsychology, such as:

  • Parisi, K.E. ; Dopp, A.R. & Quetsch, L.B. Practitioner use of and attitudes towards videoconferencing for the delivery of evidence-based telemental health interventions: A mixed methods study. Internet Interventions, 2021, 26 : 100470
  • Connolly, S. L. ; Miller, C. J. ; Lindsay, J. A. & Bauer, M. S. A systematic review of providers' attitudes toward telemental health via videoconferencing. Clinical Psychology: Science and Practice, 2020, 27(2), e12311. 10.1111/cpsp.12311
  • McKee, G.B., Pierce, B.S., Donovan, E.K., & Perrin, P.B. Examining models of psychologists' telepsychology use during the COVID‐19 pandemic: A national cross‐sectional study. Clinical Psychology, 2021, (77(10): 2405-2423.
  • Zentner, K.; Gaine, G.; Ethridge, P.; Surood, S. & Abba-Aji, A. Clinicians’ Attitudes Toward Telepsychology in Addiction and Mental Health Services, and Prediction of Postpandemic Telepsychology Uptake: Cross-sectional Study. 2022. JMIR Publications, 6(5) :e35535, doi: 10.2196/35535
  •  

Point 5: The proposed conceptual model presented at the end of the first paragraph is not explained or introduced, neither is the study itself which is also presented within the first paragraph.

Response 5: We thank you for this comment and we have modified our conceptual model (Figure 1) and we presented it in the introduction. We focused on the variables that we tested in our analysis: social telepresence and the influence of perception of telepsychology, motivation for telepsychology, training and psychologists’ characteristics. We assumed this model invariant for country and we added it in the model’s presentation in the introduction. We tested our theoretical model and we presented results related to it in part 3. (Results).

Point 6: The authors state that "According to Patel et al. [7], the COVID-19 pandemic has been associated with a marked increase in remote mental health consultation, particularly among younger patients" - please provide context, as for example the in North America, almost all mental health services, and practically all individual therapies needed to move online at certain phases of the pandemic, not simply an increase in remote mental health interventions was observed, and not only among a particular population. This was somewhat different in Europe and providing some explanation and context would be helpful, for example what determined if practices needed to more online, who made the decision etc. This may also impact the study results, in case transitioning to online was based on individual decision by the therapist, and thus pose a selection bias for the analyses in the study (i.e., including those who provided some online consultation).

Response 6: We thank you for this remark, and we chose to take off this part of the sentence, because we don’t develop it afterwards. We agree on the fact that the Covid-19 pandemic has led to an increase in publications. But if there is an increase in publications concerning professionals, many of these researches are American: it is interesting, with our article, to look elsewhere, for example on the European side. But if in Europe we are dealing with westernized countries and comparable levels of training among psychologists, there is a great cultural diversity in the familiarization with new technologies applied to the field of mental health

When we consider the situation in the countries of Europe, we are dealing not only with very diverse cultural contexts relating to the relationship between professionals and new technologies, but also with equally diverse contexts of patient care. It is important to keep this in mind, but it is equally difficult to report on it in detail in an article. This does not invalidate the value of a comparative empirical approach. Designed in a specific context with an expectation of a quick response, this type of approach confronts complex design, interpretation, and translation issues that pose unquestionable methodological problems.

The increase and diversification of research on the attitudes of mental health professionals and the acceptance of this modality justifies our contribution, but in such a vast field we have focused on specific aspects that participate in a set:

(a) online consultations, within the more general framework of telepsychology ;

(b) the role that telepresence can play in the formation of these attitudes and experiences, since it has received little attention.

Point 7: Please define telepsychiatry, especially that it seems to be used interchangeably with telepsychology.

Response 7: We thank you for your remark and we agree on this point. We added a definition of telepsychiatry in the introduction.

Point 8: The authors state "Furthermore, almost nothing is known relating to the professionals", that is, demographic and professional factors that are associated with their experiences and views on telepsychology - Actually, there is quite a few studies now published on these factors. Moreover, while it seems the authors mean teletherapy throughout the paper, they use the term "telepsychology", please clarify. Later they also use the term "teleconsultation".

Response 8: We thank you for this remark, and as we mentioned in Response 3, we tried to be clearer on our purpose and we have removed the word: "teleconsultation" from the keywords. We have also been careful to avoid confusion between the terms: "teleconsultation" and: "telepsychology" in the article. We have used the term: "telepsychology" to qualify the results we have obtained from psychologists. The term: "teleconsultation" appears only in reference to the remote consultations proposed by psychiatrists, in the framework of a medical activity.

Point 9: Methods: Sample: first 2 sentences are redundant, they were already described. It would be helpful to see the overall sample size and range by country here.

Response 9: We agree and we changed these sentences in the text. We also added data about the overall sample size in Appendix 1. We thank you for this remark.

Point 10: "The current study used a subsample of participants who provided answers to be used for a quantitative analysis " - this sounds like the original study was not quantitative, just this subsample, can you clarify?

Response 10: The large European survey of the EFPA also included open-ended questions that provided insight into experienced difficulties with online consultations. An in-depth qualitative analysis of these responses has already been published (De Witte et al., 2021) so we do not go into the qualitative data in the current report.

Point 11: "Perfect transcription of questionnaire" - I believe you meant "translation"

Response 11: We agree and we thank you for this remark. We changed: “transcription” by: “The survey was translated in 17 languages” in the text.

Point 12: Why were participants excluded from the analysis if they used another language than their national language(s) or English? How did the authors know what was the first language of the participant? This is unusual process.

Response 12: We thank you for this remark and we noticed that we made a mistake. This section in the article was based on the qualitative study that we made with the survey data (Van Daele, T.; Mathiasen, K; Carlbring, P.; Bernaerts, S.; Brugnera, A.; Compare, A.; Duque, A.; Eimontas, J.; Gosar, D.; Haddouk, L.; Karekla, M.; Larsen, P.; Lo Coco, G.; Nordgreen, T.; Salgado, J.; Schwerdtfeger, A.R.; Van Assche, E.; Willems, S.; De Witte, N.A.J. Online consultations in mental healthcare: Modelling determinants of use and experience based on an international survey study at the onset of the pandemic. Internet Interv. 2022 Sep 5;30:100571), while the current study does use the full sample. For the qualitative analysis, language was a requirement to be able to include the participants (if one of the co-authors could not understand the response, we could not analyse it of course). It doesn’t really applie for this quantitative analysis since participants could just select one of the languages and complete the multiple choice questions. We therefore cut this paragraph altogether.

Point 13: Perception of online consultations was assessed by a single question. Telepresence was defined by "feeling of being connected with one another" which does not match the commonly used definition of telepresence (see for example Bouchard et al, 2018) bar rather some aspect of a relational or alliance process. This is misleading as the study item in fact does not ask about telepresence. This was also measured by a single item.

Response 13: We thank you for this comment, we agree and we added some precisions about this point in the text: “As we aimed for brevity of the entire questionnaire, only one item was used to evaluate an aspect of telepresence, which is definitely a limitation of our method. Thus, no telepresence scale was included in our questionnaire”, which is also a limitation in the interpretation of our results.

However, we note that the question we used refers to the evaluation of "social presence" in the (VTS) (Bouchard and Robillard, 2006, 2018). (feeling that the interlocutor reacts to your presence and the impression of actively participating in exchanges with the interlocutor).”

Point 14: Results: The descriptive data in Table 1 does not describe the sample but the means and correlations between variables, therefore information about the sample demographics (age, gender, professional background etc.) is needed.

Response 14: We agree on this point, we added a description of the sample in Appendix 1.

Point 15: The results provide country comparison data that was not part of the research question.

Response 15: We mentionned in the introduction : “The current study builds on this existing work by focusing exclusively on psychologists who had adopted the practice of telepsychology, exploring the extent to which differences across countries can be retrieved.” In the results part, we provide some informations about this comparison.

Point 16: The authors phrase associations between variables by referring the impact of variable on the other, since we cannot know about causation in this cross sectional study, I suggest to refrain from such indications in general, but especially in the case of the last statement: "The results then suggest that the perception of teleconsultation has a positive impact on telepresence."

Response 16: We thank you for this remark and we agree. Given that it is a cross-sectional study, we tried to be a bit careful in our phrasing. We also added to the limitations or suggestions for further research that the associations should be confirmed in a longitudinal or experimental design.

Also, considering the limitation about the one single item and the definition of telepresence, I moderated our results and talked about : “social telepresence”.

Point 17: Discussion: The authors conclude that perception of teleconsultation is a driver of telepresence, which is not supported by the data.

Response 17: We thank you for this point, we re-phrased this part in a more tentative way and we tried to support why we think the perception of telepsychology is important.

Point 18: We know that there is a relationship between the two items (not questionnaires), and that telepresence refers more to the relationship than telepresence as understood in the previous literature, therefore this connection can be only stated very tentatively if at all.

Response 18: We thank you for this comment and we changed a bit our direction, considering only one item was used to evaluate an aspect of telepresence, which is definitely a limitation of our method, as we aimed for brevity of the entire questionnaire. We changed our results presentation, considering that we were only talking about “social telepresence”.

Reviewer 2 Report

Thanks for the paper.. seems to be very interesting... 

I have some questions for the authors...

1. The proposed “conceptual” model 1 how was tested in authors analysis? There is no model in data analysis accounted for direct or indirect effect in the regressions? We assume this model invariant for country? And if it is a conceptual theoretical model why is not tested?

Seems that some references sound needed in the paragraph "Among the concepts studied in telepsychology research, telepresence appears as the cornerstone of a successful telepractice for patient 36 and professional"

2. What type of stepwise methods was used for each Q? I mean backward forward methods? In this case it would be interesting to check for any index relative to goodness of the final model compared with the previous one. In Table 2 and 3 all the predictors are in the model even though some of them are not significant why?

3. So if you use a model assuming that "as if country were the same" what that's means in term of data analysis?

4. "To control for this variability in psychologists’ practices across countries, we mobilized a fixed-effect country model to explain perceptions of tele-practice. This model captured country specificities in a country-dummy variable."  I see that you have a sort of two level model person and country, but due to different sample size, decided to fixed country but,  What type of fixed effect model capture country specificities? I'm a little puzzled about the data showed in tables 2 and 3 and the label "Country Fixed effect"  for the parameter Country.

5. What data analysis program was used for the model estimation? And are different "econometric regression models" from the usual regression model running in research? 

6. For Q2 and Q4 where are the data that support authors conclusions? I assume but there is no cue in the text. In Q2 authors give some percentages that we do not have in any table or reference to check this data. At the same time authors said that sensitivity analysis was run without any additional control variables. What variable were in this sensitivity analysis?

7. Sorry but you need information are needed in order to justify that removing Italian subsample from data are a sort of sensitivity analysis in a regression model. As far as I know sensitivity analysis deal with variation in any of the parameter estimated while other remain unchanged.

Author Response

Point 1: The proposed “conceptual” model 1 how was tested in authors analysis? There is no model in data analysis accounted for direct or indirect effect in the regressions? We assume this model invariant for country? And if it is a conceptual theoretical model why is not tested?

Response 1: we thank you for this comment and we have modified our conceptual model (Figure 1) and we presented it in the introduction. We focused on the variables that we tested in our analysis: social telepresence and the influence of perception of telepsychology, motivation for telepsychology, training and psychologists characteristics. We assumed this model invariant for country and we added it in the model’s presentation in the introduction. We tested our theoretical model and we presented results related to it in part 3. (Results).

Point 2: Seems that some references sound needed in the paragraph "Among the concepts studied in telepsychology research, telepresence appears as the cornerstone of a successful telepractice for patient 36 and professional".

Response 2: we thank you for this point and we added some references about the feeling of telepresence and its correlation with positive emotions felt by patients in CBT videoconference psychotherapy (VCP) and the therapeutic alliance. We also mentioned tools to evaluate telepresence, such as the Videoconference Telepresence Scale (VTS) (Bouchard and Robillard, 2006, 2018).

We also specify that following your comments, we have modified the title of the article and we have also removed the word: "teleconsultation" from the keywords

Point 3: What type of stepwise methods was used for each Q? I mean backward forward methods? In this case it would be interesting to check for any index relative to goodness of the final model compared with the previous one. In Table 2 and 3 all the predictors are in the model even though some of them are not significant why?

Response 3: In this paper, we didn’t use a stepwise method because we focused on some variables of interest that we didn’t want to suppress. What we wanted to do was to emphasize the potential significance of some of them and the potential insignificance of others.

Point 4: So if you use a model assuming that "as if country were the same" what that's means in term of data analysis?

Response 4: It means that we control for specific country effect. Using a fixed effect model allow to compare results as if all country are the same.

Point 5: "To control for this variability in psychologists’ practices across countries, we mobilized a fixed-effect country model to explain perceptions of tele-practice. This model captured country specificities in a country-dummy variable."  I see that you have a sort of two level model person and country, but due to different sample size, decided to fixed country but,  What type of fixed effect model capture country specificities? I'm a little puzzled about the data showed in tables 2 and 3 and the label "Country Fixed effect"  for the parameter Country.

Response 5: We thank you for your comment and would like to point out that in many applications including econometrics and biostatistics, a fixed effects model refers to a regression model in which the group means are fixed (non-random) as opposed to a random effects model in which the group means are a random sample from a population.

REFERENCES:

Christensen, Ronald (2002). Plane Answers to Complex Questions: The Theory of Linear Models (Third ed.). New York: Springer. ISBN 0-387-95361-2.

Gujarati, Damodar N.; Porter, Dawn C. (2009). "Panel Data Regression Models". Basic Econometrics (Fifth international ed.). Boston: McGraw-Hill. pp. 591–616. ISBN 978-007-127625-2.

Hsiao, Cheng (2003). "Fixed-effects models". Analysis of Panel Data (2nd ed.). New York: Cambridge University Press. pp. 95–103. ISBN 0-521-52271-4.

Wooldridge, Jeffrey M. (2013). "Fixed Effects Estimation". Introductory Econometrics: A Modern Approach (Fifth international ed.). Mason, OH: South-Western. pp. 466–474. ISBN 978-1-111-53439-4.

Allison, Paul D. 2009. Fixed Effects Regression Models. Los Angeles: Sage Publications, Inc. (Cited 237n34.) Allstate/National Journal Heartland Monitor. 2009. ISBN 9780761924975, 0761924973

In Table 2 and in Table 3, a country dummy captures the country fixed effect. There are as many country dummies as countries.  This is why, in models presented Table 2 and table 3, there is not intercept.

Point 6: What data analysis program was used for the model estimation? And are different "econometric regression models" from the usual regression model running in research? 

Response 6: We used STATA SE16. The command used is xtreg. It can be done using R or SAS. We didn't use a stepwise method because we focus on some variables of interest that we do not want to suppress. What we want to do is to emphasize the potential significance of some of them and the potential insignificance of others.

Point 7: For Q2 and Q4 where are the data that support authors conclusions? I assume but there is no cue in the text. In Q2 authors give some percentages that we do not have in any table or reference to check this data. At the same time authors said that sensitivity analysis was run without any additional control variables. What variable were in this sensitivity analysis?

Response 7: We agree, some figures are not presented in Tables. We changed the text to make it clearer and we added Appendix 1. We thank you for this remark.

For Q2, training dimension is captured by two proxies in Table 2:

  • Telepractices before the sanitary crisis
  • Specific training

As shown Table 2, for these two proxies, the coefficients are significant. 

First, as additional preliminary statistics, we give in percentage terms, some figures on psychologists who practiced telepsychology before the sanitary crisis. Prior to the COVID-19 outbreak, the percentage of telepsychologists was respectively 58% in Spain and 54% in the UK.

For Q4, training dimension is captured by two proxies in Table 3. These two variables: Telepractices before the sanitary crisis and Specific training are significant. Therefore, we conclude as we do in this paragraph. We also added the value of coefficients in the text.

Point 8: Sorry but you need information are needed in order to justify that removing Italian subsample from data are a sort of sensitivity analysis in a regression model. As far as I know sensitivity analysis deal with variation in any of the parameter estimated while other remain unchanged.

Response 8: We thank you for this point. In the sample, the number of observations is much higher in the Italian subsample than for the other country, as we can see in Appendix 1. We deal with this issue using a fixed effect model. However, we may be interested in checking the results removing this information. It allows to see how Italian information affects the average results. We found that the main results of this paper are maintained. Therefore, we conclude that with or without the Italian subsample, the main results remain. We can complete our response by mentioning that in the scientific literature in fields as Economic, sociology, this is quite common to perform such sensitivity analysis. If here, the understanding is affected by this subsection title, we can change this title by “Robustness checks”.

Round 2

Reviewer 2 Report

Thanks for the responses to my concerns...

Thanks for the improvements made to the manuscript that make it more readable. The explanation of the analyzes is likewise more coherent with the research questions, just as there is a substantial improvement in the tables presented.

Nevertheless...

I remain thinking the figure 1. What's means  "our proposed conceptual model"... The paper states that one objective is to study the determinants of social telepresence and the role of telepsychology perception in social telepresence and refers to figure 1 as conceptual proposed model, the authors predict that psychological characteristics and level of training are factors that directly influence. However, these predictions are not answer in the paper. The authors have not conducted mediation regression analysis to test direct or indirect effect on dependant variable, and even assume that this model is invariant for country which is not tested in the paper either. Country are fixed. Authors do not tested the covariation, even the interaction between independent variables in their prediction of the dependent variable. This is what a structural model of mediation does. Therefore, figure 1, more than helping the reader, confuses the reader, mistakenly assuming that the authors have responded with their analyzes to test a model like the one in figure 1

In the response to prior revision point 3 authors argue that they did not use a stepwise methods but in the text now seems that they did. However if we check the the equations in the manuscript seems that was a Hierarchical method of entering, and if we check the table 2 and 3 seems that first entering Personal characteristics (seniority, gender) then Personal characteristics (type of employment, years of experience) and then training (Telepsychology before sanitary crisis, specific training). 

In the authors responses to prior revision, Response 6 is the same than Response 3? 

Point 3: What type of stepwise methods was used for each Q? I mean backward forward methods? In this case it would be interesting to check for any index relative to goodness of the final model compared with the previous one. In Table 2 and 3 all the predictors are in the model even though some of them are not significant why?

Response 3: In this paper, we didn’t use a stepwise method because we focused on some variables of interest that we didn’t want to suppress. What we wanted to do was to emphasize the potential significance of some of them and the potential insignificance of others.

Point 6: What data analysis program was used for the model estimation? And are different "econometric regression models" from the usual regression model running in research? 

Response 6: We used STATA SE16. The command used is xtreg. It can be done using R or SAS. We didn't use a stepwise method because we focus on some variables of interest that we do not want to suppress. What we want to do is to emphasize the potential significance of some of them and the potential insignificance of others.

Thanks!

Author Response

Point 1: I remain thinking the figure 1. What's means  "our proposed conceptual model"... The paper states that one objective is to study the determinants of social telepresence and the role of telepsychology perception in social telepresence and refers to figure 1 as conceptual proposed model, the authors predict that psychological characteristics and level of training are factors that directly influence. However, these predictions are not answer in the paper. The authors have not conducted mediation regression analysis to test direct or indirect effect on dependant variable, and even assume that this model is invariant for country which is not tested in the paper either. Country are fixed. Authors do not tested the covariation, even the interaction between independent variables in their prediction of the dependent variable. This is what a structural model of mediation does. Therefore, figure 1, more than helping the reader, confuses the reader, mistakenly assuming that the authors have responded with their analyzes to test a model like the one in figure 1

Response 1: We thank you very much for your comments. Reflecting on your comments, we finally decided that Figure 1  was not necessary and that it considered an explanatory model that justifies in particular the introduction of the bibliographical contributions in the introduction. We have therefore decided to delete Figure 1.  

Point 2: In the response to prior revision point 3 authors argue that they did not use a stepwise methods but in the text now seems that they did. However if we check the the equations in the manuscript seems that was a Hierarchical method of entering, and if we check the table 2 and 3 seems that first entering Personal characteristics (seniority, gender) then Personal characteristics (type of employment, years of experience) and then training (Telepsychology before sanitary crisis, specific training). 

Response 2: We thank you for your comment and we have tried to clarify this point by adding the following sentence to the text: “We did not use a stepwise regression model here, which involves successively adding or removing potential explanatory variables to successively test their significance. We used an incremental model, as used for example by Clark and Milcent (2018): this model seeks to examine the explanatory power of variables that are deemed essential with other groups of variables. The interest of this model, compared to the stepwise model, is not to remove non-significant variables, but to test the influence of correlations of internals to one group of variables on variables in a second group.”

Point 3: In the authors responses to prior revision, Response 6 is the same than Response 3? 

Point 3: What type of stepwise methods was used for each Q? I mean backward forward methods? In this case it would be interesting to check for any index relative to goodness of the final model compared with the previous one. In Table 2 and 3 all the predictors are in the model even though some of them are not significant why?

Response 3: In this paper, we didn’t use a stepwise method because we focused on some variables of interest that we didn’t want to suppress. What we wanted to do was to emphasize the potential significance of some of them and the potential insignificance of others.

Point 6: What data analysis program was used for the model estimation? And are different "econometric regression models" from the usual regression model running in research? 

Response 6: We used STATA SE16. The command used is xtreg. It can be done using R or SAS. We didn't use a stepwise method because we focus on some variables of interest that we do not want to suppress. What we want to do is to emphasize the potential significance of some of them and the potential insignificance of others.

Response 3: we thank you for this point and indeed, there are repetitions in the answers to points 3 and 6 of the previous remarks.

We have tried to clarify the answer we gave to the previous point 3 in point 2 of the current remarks and in the new version of the paper: “We used an incremental model, as used for example by Clark and Milcent (2018): this model seeks to examine the explanatory power of variables that are deemed essential with other groups of variables.”

The bibliographic contributions made it possible to justify that, although empirical work on the basis of surveys had indeed been carried out, it is the procedure for analyzing the variables that introduces a novelty into the explanatory model.

We have answered in point 6 regarding the data analysis program that we used for the model estimation (STATA.SE16).

We also tried to answer the following question: ” are different "econometric regression models" from the usual regression model running in research?” with the previous information that we mentioned in current Response 2 and 3. We hope that these new details will be useful and that our answers will be more satisfactory.